# Temperature effects on the calculation of the functional derivative of T$_c$ with respect to $\alpha^2 F(\omega)$

**J.A. Camargo-Martínez**[1], **F. Mesa**[2]*, **G.I. González-Pedreros**[3]

**1** Grupo de Investigación en Ciencias Básicas, Aplicación e innovación- CIBAIN, Unitrópico, Yopal, Colombia, **2** Fundación Universitaria Los Libertadores, Facultad de Ingeniería y Ciencias Básicas, Bogotá, Colombia, **3** Universidad Pedagógica y Tecnológica de Colombia, Facultad de Ciencias, Tunja, Colombia

☯ These authors contributed equally to this work.
* fredy.mesa@libertadores.edu.co

**Data Availability Statement:** All relevant data are within the paper.

**Funding:** F. Mesa was financed by the Fundación Universitaria Los Libertadores. The funders had no role in study design, data collection and analysis,

## Abstract

The functional derivative of the superconducting transition temperature T$_c$ with respect to the electron-phonon coupling function $\alpha^2 F(\omega)$, $\delta T_c^2 / \delta\alpha^2 F(\omega)$ permits identifying the frequency regions where phonons are most effective in raising T$_c$. This work presents an analysis of temperature effects on the calculation of the $\delta T_c / \delta\alpha^2 F(\omega)$ and $\mu^*$ parameters. The results may permit establishing that the variation of the temperature in the $\delta T_c / \delta\alpha^2 F(\omega)$ and $\mu^*$ parameter allows establishing patterns and conditions that are possibly related to the physical conditions in the superconducting state, with implications on the theoretical estimation of the T$_c$.

## Introduction

Superconductivity is the complete loss of electrical resistivity of a material that occurs only below a certain temperature, called superconducting critical temperature T$_c$. It is a state of matter with technologically impactful applications but with serious difficulties of use on a large scale due to the extreme conditions in which it occurs: low temperatures or high pressures. However, its application on a small scale is a current fact.

Research on the subject from a theoretical approach seeks to establish its fundamental physical mechanisms, the understanding of which will positively lead to the engineering of superconducting materials with a view to their application on a large scale. The best approach, recognized with the Nobel Prize in physics in 1971, is the Bardeen-Cooper-Schrieffer (BCS) theory, which states that superconductivity is the "physics of Cooper pairs" [1].

Here, the effective attraction between electrons forming the Cooper pair is generated by the interaction between the electrons and the lattice vibrations (phonons), called the electron-phonon interaction. This scheme explains the phenomenon for weak electron-phonon coupling systems, leading to T$_c$ below 70 K, (lower than the temperature of liquid nitrogen). Thus, the next step was to generalize the BCS theory to superconductors, in which the electron-phonon interaction is strong and hence has a higher T$_c$ This was the work of G. M. Eliashberg [2] who in his theoretical description, introduced the electron-phonon interaction and the electronic

decision to publish, or preparation of the manuscript.

**Competing interests:** The authors have declared that no competing interests exist.

and phononic band structure more precisely. All that information is gathered in a function, the Eliashberg spectral function, or electron-phonon coupling function $\alpha^2 F(\omega)$ (see Fig 1), which can be obtained both theoretically (DFT calculations) and experimentally (tunneling experiment). The Eliashberg spectral function is obtained from the calculated phonon spectrum and the calculated electron–phonon matrix elements [3, 4]. The Coulombic repulsion between electrons is included through a parameter $\mu$.

On the other hand, the linearization of the Eliashberg equations makes it possible to determine the functional derivative of the superconducting critical temperature $T_c$ concerning the function $\alpha^2 F(\omega)$, $\delta F_c/\delta\alpha^2 F(\omega)$. The first numerical calculations of the $\delta F_c/\delta\alpha^2 F(\omega)$ in superconductors were performed by Bergmann and Rainer [5]. Their results showed that this function has a universal form (see Fig 1b): it grows from $\omega = 0$ to a maximum at $\omega \sim 7K_B T_c$ and then slowly decreases to 0 as $\omega \to \infty$ [5, 6]. From $\delta T_c/\delta\alpha^2 F(\omega)$, it is possible to determine the phonon frequency leading to the highest possible $T_c$ in a superconductor [7] and to describe the change in $T_c$, $\Delta T_c$, given a slight variation in the $\alpha^2 F(\omega)$ function, $\Delta\alpha^2 F(\omega)$, generated by the action of physical conditions such as pressure, doping, etc. [8–10], thus (Eq (1);

$$\Delta T = \int_0^{+\infty} \frac{\Delta T_c}{\delta\alpha^2 F(\omega)} \Delta\alpha^2 F(\omega) d\omega \tag{1}$$

A previous theoretical study showed that there is a correlation between the frequencies of the maxima of the $\delta T_c/\delta\alpha^2 F(\omega)$ and $\alpha^2 F(\omega)$ functions [11, 12], where the convergence of these frequencies occurs at the optimal electron-phonon interaction conditions leading to the superconductor reaching the maximum possible $T_c (T_c^{Max})$ [13–17]. This convergence frequency is called the optimum frequency $\omega_{opt}$, which satisfies the relation $\omega_{opt} = 7K_B T_c^{Max}$, where $K_B$ is the Boltzmann constant. The calculation of the $\delta T_c/\delta\alpha^2 F(\omega)$ requires the experimental (or test) $T_c$ value for the prior determination of the parameter $\mu^*$, which physically accounts for the Coulombic repulsion between electrons in the system under study. The parameter $\mu^*$ is determined from the fit to the linearized Eliashberg equations (see Materials and methods section-Eq (3) when the pair breakdown parameter $\rho$ tends to zero ($\rho \to 0$), which is valid for $T = T_c$ [15].

Up until now, the physical interpretation and application of the $\delta T_c/\delta\alpha^2 F(\omega)$ have failed to consolidate. In 2015 Nicol and Carbotte used the $\delta T_c/\delta\alpha^2 F(\omega)$ to demonstrate that the $\alpha^2 F(\omega)$

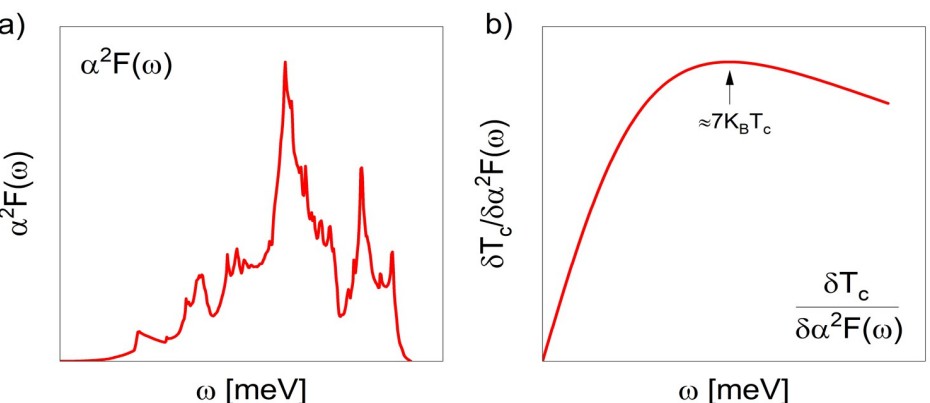

**Fig 1.** Schematics of (a) the Eliashberg spectral function $\alpha^2 F(\omega)$ and (b) the functional derivative of the superconducting critical temperature $T_c$ concerning the $\alpha^2 F(\omega)$ function, $\Delta T_c/\delta\alpha^2 F(\omega)$.

spectral function of sulfur trihydride $H_3S$ at 200 GPa is highly optimized for $T_c$ [18]. González-Pedreros and Baquero [10] and Camargo-Martínez *et al.* [19] used the $\delta T_c/\delta\alpha^2F(\omega)$ to determine the trend of $T_c$ as a function of pressure in Nb-bcc (Cubic Niobium) and $H_3S$ respectively, taking the reported experimental $T_c$ as a starting point. In other work, the $\delta T_c/\delta\alpha^2F(\omega)$ was determined to identify possible frequency regions where phonons would be the most effective in increasing $T_c$ [20, 21]. All these results are descriptive and not predictive in nature.

One of the possible contributions of theoretical physics in superconductivity is to clearly establish the fundamental physical foundations of the superconducting phenomenon in order to suggest with certainty, the line of experimental process to obtain superconductivity at room temperature in viable conditions for its application to large-scale. An example of the predictive effect of the theoretical approach on superconductivity was observed in the idea proposed by Ashcroft [22], who stated that hydrogen-rich systems would be viable candidates to be high critical temperature superconductors. This proposal gave rise to experimentation in this field with the discovery of new high-Tc superconductors, as $H_3S$ ($T_c$ of 203 K at 155 GPa [23]) or $LaH_{10}$ ($T_c$ of 260 K at 180 GPa [24]), called hydride superconductors. This discovery gave a new impetus to this field of study, which had been stuck with the superconducting cuprates ($T_c$ of 164 K) since 1994 [25]. The current difficulty with hydride superconductors is in their high-pressure conditions of formation. In this sense, evaluating possible new ways to predict $T_c$ values in terms of well-defined physical conditions (such as pressure, doping, etc.) is an interesting line of work. Here, the study of the functional derivative $\delta T_c/\delta\alpha^2F(\omega)$ seeks to establish the possible existence of patterns that lead to the determination of an optimum temperature of the system (superconducting critical temperature), which would also avoid the use of test or experimental $T_c$ in first-principles calculations.

From a purely computational point of view, the temperature value can have pivotal implications in the calculation, result, and interpretation of the functional derivative $\delta T_c/\delta\alpha^2F(\omega)$. For this reason, in this manuscript, we present the analysis of the effects of temperature variation, around experimental $T_c$ value, on the calculation of the functional derivative $\delta T_c/\delta\alpha^2F(\omega)$, in the superconductor $H_3S$, of which a $T_c$ of 203 K at 155 GPa was measured [23].

## Materials and methods

This study was developed based on the Eliashberg $\alpha^2F(\omega)$ spectral functions of $H_3S$ obtained in previous work [12, 19], whose calculations were performed in the range of pressures (155 – 225 GPa), where the experimental $T_c$ were reported [23]. Here, the functional derivatives were obtained with the procedure widely used by Carbotte *et al.* [13–16, 18, 26, 27] which is based on the work of G. Bergmann and D. Rainer [5]. The determination of the functional derivative (see Eq (2)) of the superconducting critical temperature $T_c$ with respect $\alpha^2F(\omega)$ function, $\delta T_c/\delta\alpha^2F(\omega)$, was performed from the relation:

$$\frac{\delta T_c}{\delta\alpha^2F(\omega)} = \frac{\dfrac{\delta\rho}{\delta\alpha^2F(\omega)}}{\left(\dfrac{\partial\rho}{\partial T}\right)_{T_c}} \tag{2}$$

Where $\rho$ was expressed in terms of $K_{nm}$ for $T = T_c$, which is obtained as a (kernel) solution of the linearized Eliashberg equations on the imaginary axis [5, 6]:

$$\rho\overline{\Delta}_n = \pi T\sum_m\left(\lambda_{nm} - \mu^* - \delta_{nm}\frac{|\tilde{\omega}|}{\pi T}\right)\overline{\Delta}_m \tag{3}$$

with $\tilde{\omega} = i\omega_n + \pi T\sum_m\lambda_{nm}sgn(\omega_n)$, $\omega_n = \pi T(2n - 1)$ the n-th Matsubara frequency and

$\overline{\Delta}_n = \frac{\tilde{\Delta}_n}{\rho + |\tilde{\omega}_n|}$. For more details on the mathematical formulation, see the reference [17]. To evaluate the effects of the $T_c$ parameter on the $\delta T_c/\delta\alpha^2 F(\omega)$ calculations, 10 K variations in temperature around the experimental value (reference temperature) were developed for each of the pressures evaluated (155 GPa,175 GPa, 195 GPa, and 215 GPa).

## Results and discussion

The $\delta T_c/\delta\alpha^2 F(\omega)$ as a function of frequency calculated for $H_3S$ at different pressures and temperatures are presented in Fig 2.

Each case evaluated (Fig 2), frequency values in every $\delta T_c/\delta\alpha T2F(\omega)$-maximum are displaced, $\delta_{MAX}$, as temperature is increased in accord with relation $\omega_{opt} \sim k_B T_c$ [5]. The $\delta T_c/\delta\alpha^2 F(\omega)$-maximum allows identifying of the frequency regions $\omega_{opt}$ where phonons are more effective in increasing $T_c$ [28]; this is why it is so important to evaluate their behavior.

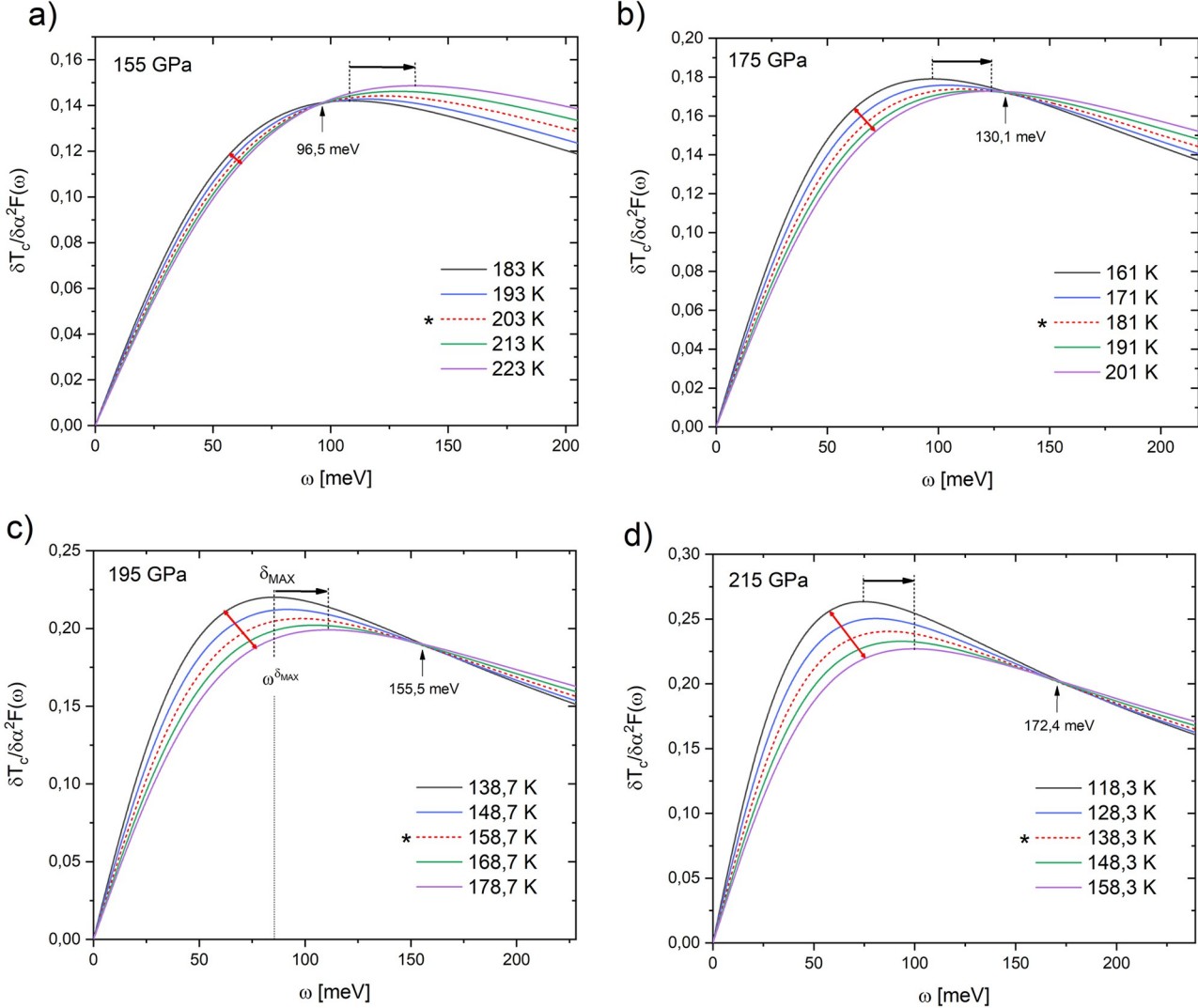

**Fig 2. $\delta T_c/\delta\alpha^2 F(\omega)$ as a function of frequency $\omega$ for $H_3S$ calculated at different temperatures at pressures of a) 155GPa, b) 175 GPa, c) 195 GPa, and d) 215 GPa.** Stars show critical temperature related to corresponding pressure, horizontal arrows indicate frequency shift of the maximum value of the $\delta T_c/\delta\alpha^2 F(\omega)$, $\omega^{\delta_{MAX}}$, red arrows present how $\delta T_c/\delta\alpha^2 F(\omega)$-curve increases their separation, and vertical arrows point average frequency of intersection of the $\delta T_c/\delta\alpha^2 F(\omega)$.

To every pressure (Fig 2a–2d), in a temperature interval $\Delta T = 40K$ about $T_c$, $\delta T_c/\delta \alpha^2 F(\omega)$ intersects a narrow frequency range (it looks point-like) is verified. Condition of small variation of the $\delta T_c/\delta \alpha^2 F(\omega)$ due to temperature effects, below the intersection point, is observed for the system under the lowest compression (155 GPa) and vice versa. This behavior is opposite if the $\delta T_c/\delta \alpha^2 F(\omega)$ are compared at frequencies higher than the intersection point. In both cases, the effect of temperature on the calculation of the $\delta T_c/\delta \alpha^2 F(\omega)$ could establish patterns (intersection point, variation, or separation between the $\delta T_c/\delta \alpha^2 F(\omega)$ and their maxima) that would lead to the determination of optimal physical conditions of the superconducting state and the possible estimation of the $T_c$.

Now, the intensities (value on the vertical axis) of the $\delta T_c/\delta \alpha^2 F(\omega)$-maximums show two different behaviors (Fig 2). At 155 GPa, such intensities slightly increase their value as the temperature increases, as a consequence of the little separation induced in the $\delta T_c/\delta \alpha^2 F(\omega)$. However, for the other pressures (175, 195, and 215 GPa), the behavior of the $\delta T_c/\delta \alpha^2 F(\omega)$ intensities is opposite to the 155 GPa case, starting from a higher intensity and decreasing with increasing temperature, being more evident with increasing pressure. It is important to note that there seems to be no relationship between the variation of the frequency of the maximum of the $\delta T_c/\delta \alpha^2 F(\omega)$, $\omega^{\delta MAX}$, and the intensity of the maximum of the $\delta T_c/\delta \alpha^2 F(\omega)$.

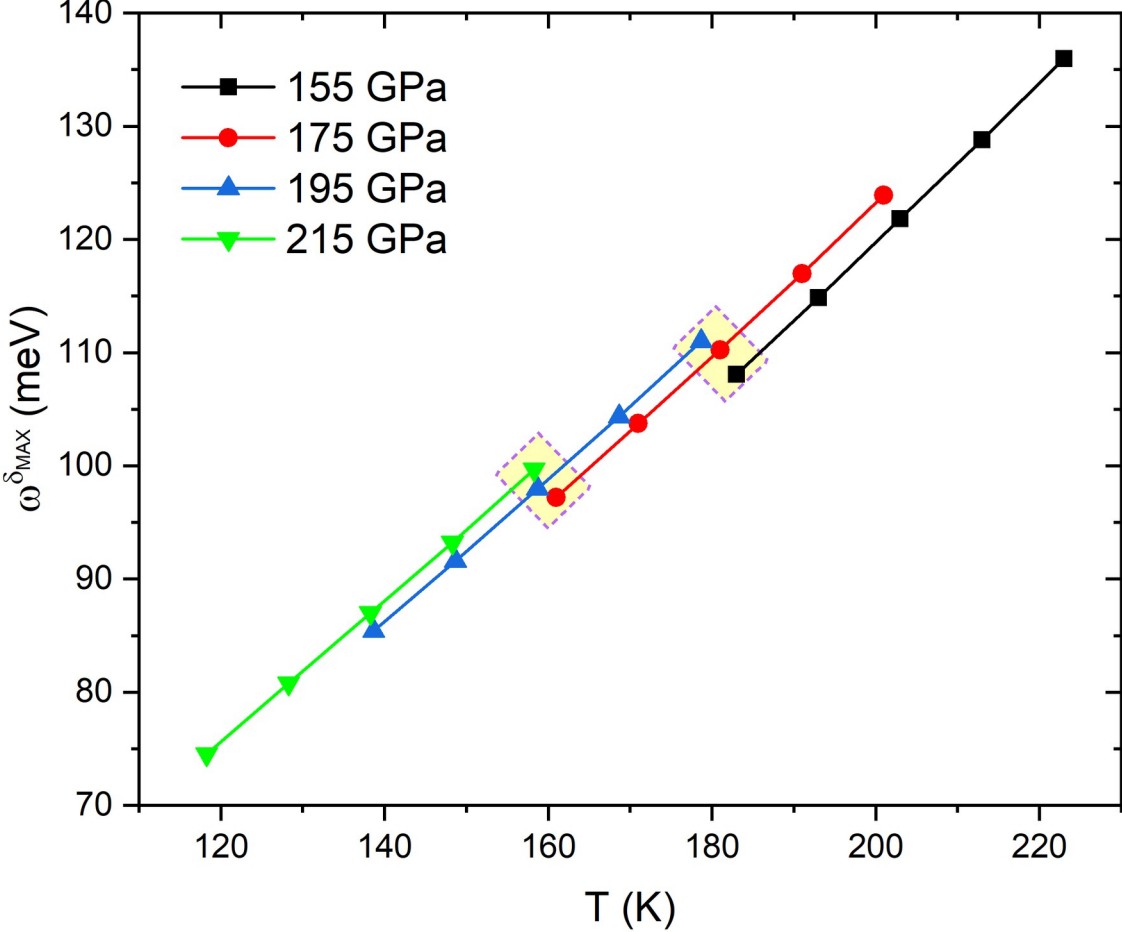

**Fig 3. Frequency of the maximum of the $\delta T_c/\delta \alpha^2 F(\omega)$, $\omega^{\delta MAX}$, as a function of temperature, at different pressures.** The dashed boxes (yellow) show the slight change in $\omega^{\delta MAX}$ induced by pressure.

The patterns of the $\delta T_c/\delta\alpha^2 F(\omega)$ vs $T_c$ in the $H_3S$ reveal that these seem to have a characteristic behavior at a specific pressure (155 GPa).

(Fig 3) shows the linearity of the frequency of the maximum of the $\delta T_c/\delta\alpha^2 F(\omega)(\omega^{\delta MAX})$ as a function of temperature for all pressures. Such lines are collinear with mean slope $\overline{m} = +0.64$ meV/K. This means that the $\omega^{\delta MAX}$ moves uniformly toward higher frequencies as the temperature increases. On the other hand, $\omega^{\delta MAX}$ is almost unaffected by the pressure (p) since a considerable change of $\Delta p = 40$ GPa induces a small $\Delta\omega^{\delta MAX} = 2.5$ meV. However, each pressure has a limit of $\omega^{\delta MAX}$, whose maximum value is reached at 155 GPa, leading to a higher $T_c$.

In the calculation of the $\delta T_c/\delta\alpha^2 F(\omega)$, the temperature variation involves the determination of the $\mu^*$ parameter of Fig 3. The comparison between the parameters $\mu^*$ adjusted at different temperatures for each of the pressures is presented in Fig 4.

It is observed in Fig 4 that $\mu^*$ vs T has a comparable trend between pressures. This could be assumed to be quasi-linear in a first approximation. However, this quasi-linearity is much more evident at higher pressure. The results show that a $\Delta T_c = 40K$ induces a $\overline{\Delta\mu^*} = 0.25$. However, the $\mu^*$ values fitted to the experimental $T_c$ in the pressure range from 155 to 215 GPa are in the $\Delta\mu^*$ range of 0.209–0.285, which implies a $\overline{\Delta T_c} = 18.1$ K, a small range with respect

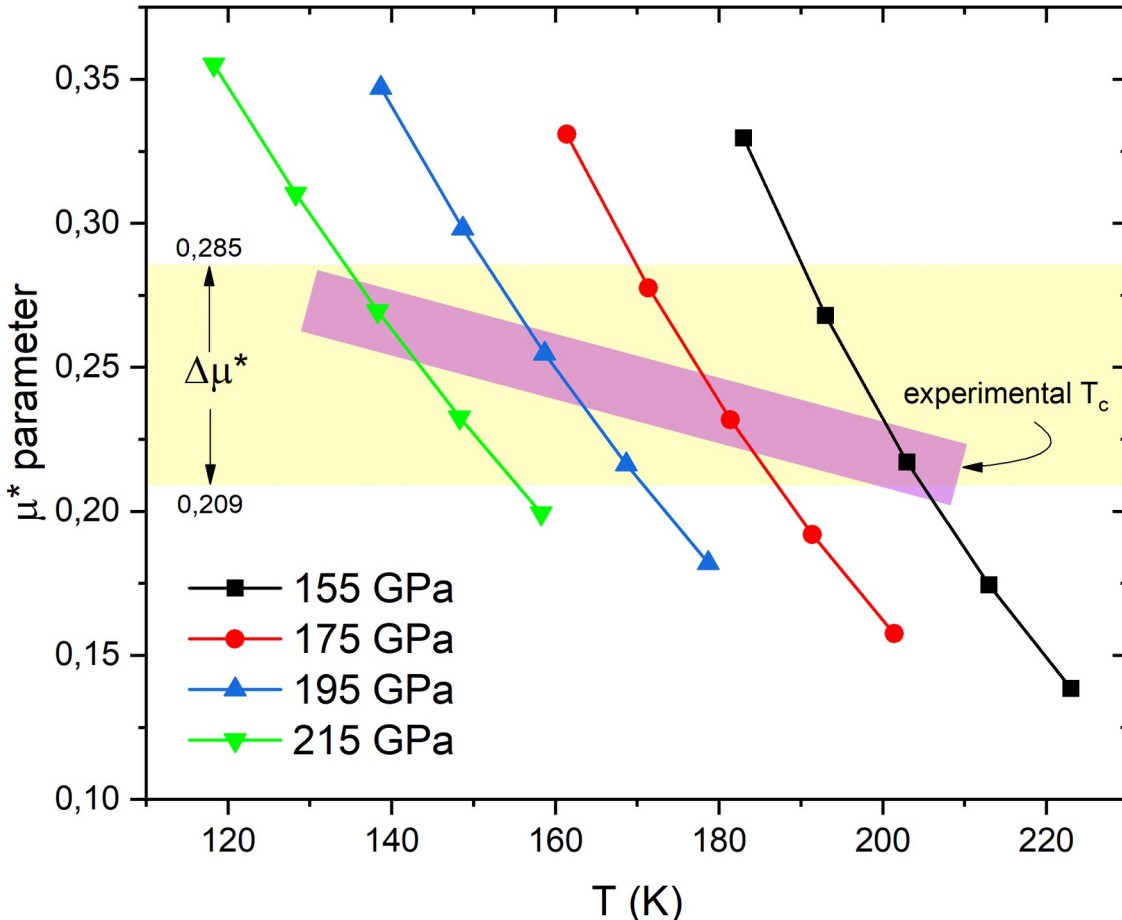

**Fig 4. The parameter $\mu^*$ as a function of temperature T for $H_3S$ at different pressures.** The horizontal band (yellow) marks the $\Delta\mu^*$ within which the $\mu^*$ fitted to the experimental $T_c$ are set, indicated by the diagonal band (purple).

to the 203 K of the experimental maximum $T_c$ of $H_3S$. This result is interesting because it would allow establishing an initial criterion of theoretical estimation of the $T_c$ around a small range of temperatures, according to the $\Delta\mu^*$. It is then necessary to determine, with this procedure, the $\Delta\mu^*$ other systems to evaluate if this presents a universal range or if it varies significantly from one system to another.

In the calculation of the $\delta T_c/\delta\alpha^2 F(\omega)$, it was found that for temperatures distant between -60 K and +30 K with respect to the experimental $T_c$, $\mu^*$ values of 0,8 and 0,09 are generated, which are outside the values typically used or calculated (between 0,3 and 0,1), and their $\delta T_c/\delta\alpha^2 F(\omega)$ presented computational difficulties in their calculation, with behaviors different in form from those observed in the $\delta T_c/\delta\alpha^2 F(\omega)$ calculated at temperatures close to the experimental $T_c$.

## Conclusions

This paper presents the preliminary theoretical analysis of the effects of temperature variation around the experimental superconducting critical temperature $T_c$ on the calculation of the functional derivative $\delta T_c/\delta\alpha^2 F(\omega)$ for superconducting $H_3S$ in the pressure range from 155 to 215 GPa. These calculations included the determination of the $\mu^*$ parameters through fitting $T_c$ in the linearized Eliashberg equations. The calculated $\delta T_c/\delta\alpha^2 F(\omega)$ revealed temperature- and pressure-induced displacement, intersection, and separation patterns that could be associated with the physical conditions in the superconducting state and the estimation of $T_c$. The $\mu^*$ values obtained allowed the determination of a range of values leading to temperatures that could establish an initial criterion for possible theoretical estimation of $T_c$. This procedure must be evaluated and confirmed in other similar systems to establish the possible generalization of the results presented here.

## Author Contributions

**Conceptualization:** J.A. Camargo-Martínez.

**Data curation:** J.A. Camargo-Martínez, F. Mesa, G.I. González-Pedreros.

**Formal analysis:** J.A. Camargo-Martínez, G.I. González-Pedreros.

**Investigation:** J.A. Camargo-Martínez, F. Mesa, G.I. González-Pedreros.

**Methodology:** J.A. Camargo-Martínez, G.I. González-Pedreros.

**Supervision:** F. Mesa.

**Validation:** J.A. Camargo-Martínez, F. Mesa, G.I. González-Pedreros.

**Visualization:** F. Mesa.

**Writing – original draft:** J.A. Camargo-Martínez, F. Mesa, G.I. González-Pedreros.

**Writing – review & editing:** F. Mesa.

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
