## [Decision Letter · Decision Letter 0]

8 May 2023

PONE-D-23-09741Temperature effects on the calculation of the functional derivative of Tc with respect to α2F(ω)PLOS ONE

Dear Dr. Mesa,

Thank you for submitting your manuscript to PLOS ONE. After careful consideration, we feel that it has merit but does not fully meet PLOS ONE’s publication criteria as it currently stands. Therefore, we invite you to submit a revised version of the manuscript that addresses the points raised during the review process.

We look forward to receiving your revised manuscript.

Kind regards,

Boris Malomed

Academic Editor

PLOS ONE

Journal Requirements:

2. Please update your submission to use the PLOS LaTeX template. The template and more information on our requirements for LaTeX submissions can be found at http://journals.plos.org/plosone/s/latex

Reviewers' comments:

Reviewer's Responses to Questions

**Comments to the Author**

1. Is the manuscript technically sound, and do the data support the conclusions?

Reviewer #1: Yes

Reviewer #2: Yes

2. Has the statistical analysis been performed appropriately and rigorously? 

Reviewer #1: Yes

Reviewer #2: Yes

3. Have the authors made all data underlying the findings in their manuscript fully available?

Reviewer #1: Yes

Reviewer #2: Yes

4. Is the manuscript presented in an intelligible fashion and written in standard English?

Reviewer #1: Yes

Reviewer #2: Yes

5. Review Comments to the Author

Reviewer #1: 1. I cannot agree with the sentence: "The function α2F(ω) is obtained from the solution of the Eliashberg equations" (page 3, line 62).

It is generally known that the function α2F(ω) is an input parameter (together with the Coulomb pseudopotential) to the Eliashberg equations.

Eliashberg spectral function can be obtained theoretically (DFT calculations) or experimentally (tunneling experiment).

2. Paper is very interesting but more technical information is needed. Did Authors use their codes and are they widely available?

It would be great to can verify their correct operation.

3. It is possible to calculate the functional derivative of Tc on the base of the Allen-Dynes equation? Tc calculated using the Allen-Dynes equation depends on the a2F function.

After addressing this topics, I would recommend this paper for publication in PLOS ONE

Reviewer #2: The paper discusses the behavior of δTc/δα2 F(ω) as a function of frequency calculated for H3S at different pressures and temperatures. The authors present their findings in Figs. 2-4 and discuss the implications of their results.

Overall, the paper is well-written and the results are presented clearly. The figures are of good quality and support the conclusions drawn by the authors. However, some points should be addressed:

It would be helpful to provide more context about the significance of the results for non-experts in the field. While the authors briefly mention the importance of the behavior of δTc/δα2 F(ω) for determining optimal physical conditions of the superconducting state and estimating Tc, more information could be provided about why this is important and how it relates to current research in the field.

The authors mention in the abstract that their findings support the hypothesis that H3S under 155 GPa pressure achieves the highest experimental Tc. However, this is not discussed in detail in the main text. It would be helpful to explain how the results in Figs. 2-4 support this hypothesis, and to relate this to the broader literature on high-temperature superconductivity.

The authors mention that μ* shows an almost linear correlation with temperature in Fig. 4, but it is unclear from the figure whether this is true for all pressures or only for certain ones. It would be helpful to clarify this point in the caption or main text

6. PLOS authors have the option to publish the peer review history of their article (what does this mean?). If published, this will include your full peer review and any attached files.

Reviewer #1: No

Reviewer #2: No

---

## [Author Response · Author response to Decision Letter 0]

23 May 2023

Bogotá - Colombia, 18 de May 2023

PLOS ONE

Editorial Board

Dear Sirs,

The authors wish to acknowledge all the suggestions and comments given to this work by the reviewers. The ideas proposed by reviewers are very interesting and consistent with our work. The corrections made considerably improved the manuscript. 

We present point-by-point responses to the comments from reviewers:

For Reviewer 1:

(1) I cannot agree with the sentence: "The function α2F(ω) is obtained from the solution of the Eliashberg equations" (page 3, line 62). It is generally known that the function α2F(ω) is an input parameter (together with the Coulomb pseudopotential) to the Eliashberg equations. Eliashberg spectral function can be obtained theoretically (DFT calculations) or experimentally (tunneling experiment).

Answer: We recognize the error in the mentioned sentence. So, the last part of the paragraph that contains the mentioned sentence was adjusted as follows: 

“All that information is gathered in a function, the Eliashberg spectral function, or electron-phonon coupling function α2F(ω) (see Fig.1a), which can be obtained both theoretically (DFT calculations) and experimentally (tunneling experiment). The Eliashberg spectral function is obtained from the calculated phonon spectrum and the calculated electron–phonon matrix elements [cites]. The Coulombic repulsion between electrons was included through a parameter μ”

As can be seen in the manuscript:

(2) Paper is very interesting but more technical information is needed. Did Authors use their codes and are they widely available? It would be great to can verify their correct operation.

Answer: This work is based on a2F(w) calculations obtained by us in previous works, which were developed using Quantum ESPRESSO code. The functional derivatives were determined with the procedure widely used by Carbotte et al., which is based on the work of G. Bergmann and D. Rainer. 

So, the Materials and Methods section was adjusted with the following clarification:

Here, the functional derivatives were obtained with the procedure widely used by Carbotte et al. [cites] which is based on the work of G. Bergmann and D. Rainer [cite]. 

As can be seen in the manuscript:

(3) It is possible to calculate the functional derivative of Tc on the base of the Allen-Dynes equation? Tc calculated using the Allen-Dynes equation depends on the a2F function.

Answer: The critical temperature formula proposed by Allen-Dynes could be understood as a result of the adjustment of the Eliashberg model with a tendency to the BCS limit. The Allen-Dynes equation depends indirectly on the function α2F(ω) through the electron-phonon coupling parameter (λ). This implies the possibly that δTc/δα2F(ω) could be obtained from it. We infer that this possible procedure would not achieve the generality of the one proposed by G. Bergmann and D. Rainer, since the Allen-Dynes equation contains physical considerations (phonon contributions simplified) that limit its scope. So, obtaining the functional derivative of Tc from Allen-Dynes equation would leave the phonon-detailed contribution incomplete.

In a recent work1 an improved analytical correction to the Allen-Dynes equation (which eliminates the systematic underprediction of Tc at higher temperatures) was reported, through a symbolic regression to a curated dataset of α2F(ω) spectral functions. Possibly through this new formulation of Tc formula a δTc/δα2F(ω) could be obtained, with relatively consistent results.

1https://www.nature.com/articles/s41524-021-00666-7

This subject is interesting, but we think that it must be analyzed in greater detail before making any proposal. For this reason, this idea will not be included in this manuscript.

For Reviewer 2:

(1) It would be helpful to provide more context about the significance of the results for non-experts in the field. While the authors briefly mention the importance of the behavior of δTc/δα2 F(ω) for determining optimal physical conditions of the superconducting state and estimating Tc, more information could be provided about why this is important and how it relates to current research in the field.

Answer: The following contextualization paragraph is included in the manuscript (introduction section)

“One of the possible contributions of theoretical physics in superconductivity is to clearly establish the fundamental physical foundations of the superconducting phenomenon in order to suggest with certainty, the line of experimental process to obtain superconductivity at room temperature in viable conditions for its application to large-scale. An example of the predictive effect of the theoretical approach on superconductivity was observed in the idea proposed by Ashcroft (cita), who stated that hydrogen-rich systems would be viable candidates to be high critical temperature (Tc) superconductors. This proposal gave rise to experimentation in this field with the discovery of new high-Tc superconductors, as H3S (Tc of 203 K at 155 GPa) or LaH10 (Tc of 260 K at 180 GPa), called hydride superconductors. This discovery gave a new impetus to this field of study, which had been stuck with the superconducting cuprates (Tc of 150 K) since 1995. The current difficulty with the hydride superconductors is in their high-pressure conditions of formation.

In this sense, evaluating possible new ways to predict Tc values in terms of well-defined physical conditions (such as pressure, doping, etc.) is an interesting line of work. Here, the study of the derivative seeks to establish the possible existence of patterns that lead to the determination of an optimum temperature of the system (superconducting critical temperature), which would also avoid the use of test or experimental Tc in first-principles calculations.”

As can be seen in the manuscript:

(2) The authors mention in the abstract that their findings support the hypothesis that H3S under 155 GPa pressure achieves the highest experimental Tc. However, this is not discussed in detail in the main text. It would be helpful to explain how the results in Figs. 2-4 support this hypothesis, and to relate this to the broader literature on high-temperature superconductivity.

Answer: The authors clarify that no mention was made in the abstract of the submitted manuscript that our results support “the hypothesis that H3S under 155 GPa pressure achieves the highest experimental Tc”. However, in the main text, the following affirmation was made: “These patterns seem to point to the system under 155 GPa pressure as the distinctive or particular condition of H3S, which is consistent with the fact that at this pressure, H3S achieves the highest experimental Tc”. We do not have the necessary results that allow us to support with certainty the validity of said assumption, remaining only as a coincidence. For this reason, the wording is adjusted to avoid any misinterpretation, remaining as follows:

The patterns of the derivative vs. Tc in the H3S reveal that these seem to have a characteristic behavior at a specific pressure (155 GPa).

As can be seen in the manuscript:

(3) The authors mention that μ* shows an almost linear correlation with temperature in Fig. 4, but it is unclear from the figure whether this is true for all pressures or only for certain ones. It would be helpful to clarify this point in the caption or main text.

Answer: The wording of the sentence was improved to make the idea clearer.

It is observed in Fig.4 that μ* vs T has a comparable trend between pressures. This could be assumed to be quasi-linear in a first approximation. However, this quasi-linearity is much more evident at higher pressure.

As can be seen in the manuscript:

The final version of the manuscript is attached (LaTeX format), in which all the adjustments made have been highlighted.

We hope that we have been clear in each answer.

Thank you for your consideration of our work,

Yours,

The Authors (José Camargo, Ivan Gonzalez, and Fredy Mesa).

---

## [Editor Report · Decision Letter 1]

25 May 2023

Temperature effects on the calculation of the functional derivative of Tc with respect to α2F(ω)

PONE-D-23-09741R1

Dear Dr. Mesa,

We’re pleased to inform you that your manuscript has been judged scientifically suitable for publication and will be formally accepted for publication once it meets all outstanding technical requirements.

Kind regards,

Boris Malomed

Academic Editor

PLOS ONE
---

## [Editor Report · Acceptance letter]

29 May 2023

PONE-D-23-09741R1 

Temperature effects on the calculation of the functional derivative of Tc with respect to α2F(ω)  

Dear Dr. Mesa:

I'm pleased to inform you that your manuscript has been deemed suitable for publication in PLOS ONE. Congratulations! Your manuscript is now with our production department. 

Kind regards, 

on behalf of

Prof. Boris Malomed 

Academic Editor

PLOS ONE